# Neuroprotective Effects of *Ecklonia cava* in a Chronic Neuroinflammatory Disease Model

**DOI:** 10.3390/nu15082007

**Published:** 2023-04-21

**Authors:** Seong-Lae Jo, Hyun Yang, Kang-Joo Jeong, Hye-Won Lee, Eui-Ju Hong

**Affiliations:** 1College of Veterinary Medicine, Chungnam National University, Daejeon 34134, Republic of Korea; jsr7093@nate.com (S.-L.J.); rkdwn87@naver.com (K.-J.J.); 2KM Convergence Research Division, Korea Institute of Oriental Medicine, Daejeon 34054, Republic of Korea; hyunyang@kiom.re.kr

**Keywords:** neuroinflammation, *Ecklonia cava*, neurodegenerative disease

## Abstract

Inflammation is a natural defense mechanism against noxious stimuli, but chronic inflammation can lead to various chronic diseases. Neuroinflammation in the central nervous system plays an important role in the development and progression of neurodegenerative diseases. Polyphenol-rich natural products, such as *Ecklonia cava* (*E. cava*), are known to have anti-inflammatory and antioxidant properties and can provide treatment strategies for neurodegenerative diseases by controlling neuroinflammation. We investigated the effects of an *E. cava* extract on neuroinflammation and neurodegeneration under chronic inflammatory conditions. Mice were pretreated with *E. cava* extract for 19 days and then exposed to *E. cava* with lipopolysaccharide (LPS) for 1 week. We monitored pro-inflammatory cytokines levels in the serum, inflammation-related markers, and neurodegenerative markers using Western blotting and qRT-PCR in the mouse cerebrum and hippocampus. *E. cava* reduced pro-inflammatory cytokine levels in the blood and brain of mice with LPS-induced chronic inflammation. We also measured the activity of genes related to neuroinflammation and neurodegeneration. Surprisingly, *E. cava* decreased the activity of markers associated with inflammation (NF-kB and STAT3) and a neurodegenerative disease marker (glial fibrillary acidic protein, beta-amyloid) in the cerebrum and hippocampus of mice. We suggest that *E. cava* extract has the potential as a protective agent against neuroinflammation and neurodegenerative diseases.

## 1. Introduction

Inflammation is a biological defense mechanism against harmful stimuli from injuries or infections [1]. Inflammation can be triggered by various factors, including infections, environmental factors, lifestyle choices, and underlying genetic predispositions [2]. The function of inflammation is to prevent the initial cause of cell damage, remove destroyed tissue and necrotic cells from the wound, and repair the tissue [3]. Inflammation can be acute or chronic. Acute inflammation causes immediate injury lasting from a few days to a few weeks [4]. Cytokines and chemokines promote the migration of neutrophils and macrophages to areas of inflammation. Persistent acute inflammation can lead to chronic inflammation [5].

Chronic inflammation can last months or years [1,5]. Chronic inflammation is caused by the persistence of a harmful trigger and a continuous inflammatory response [1]. Chronic inflammation releases pro-inflammatory cytokines and other signaling molecules that disrupt normal physiological functions of the immune system [1]. The constant occurrence of these inflammatory processes can lead to the activation of immune cells such as macrophages and T cells, which can increase the inflammatory response and lead to tissue damage and dysfunction [6]. Based on this biological damage, chronic inflammation is known to contribute to the development of chronic diseases by promoting oxidative stress, DNA damage, and cellular dysfunction [7,8,9]. These chronic diseases such as cardiovascular diseases, type 2 diabetes, cancer, autoimmune disorders, and neurodegenerative diseases [2,10].

Surprisingly, diabetes, vascular disorders, and heart disease are associated with neuroinflammation [11,12,13,14], which plays a crucial role in the development and progression of various neurodegenerative diseases, such as Alzheimer’s disease and Parkinson’s disease [10,15]. Neuroinflammation (NI) is a process of inflammation that occurs in the central nervous system (CNS) as a response to injury, infection, traumatic brain injury, toxic metabolites, or autoimmunity [16]. This process involves the activation of immune cells such as microglia and astrocytes, and the release of pro-inflammatory cytokines, chemokines, and other signaling molecules [17]. The release of inflammatory molecules can cause tissue damage and cell death, which in turn contribute to the development and progression of neurodegenerative diseases [18,19]. Therefore, NI plays a critical role in conditions such as Alzheimer’s and Parkinson’s diseases [10,15]. Given that NI has been linked to numerous neurodegenerative diseases (ND), there is growing interest in whether alleviating inflammation could potentially lead to a reversal of neurodegeneration [20].

Recent studies demonstrated that natural products can relieve NI, thereby providing a therapeutic strategy for ND [21,22,23]. These products include flavonoids, phlorotannins, curcumin, resveratrol, and genistein [23,24,25,26,27]. *Ecklonia cava*, a brown alga commonly found in the coastal waters of East Asia, has been used in traditional medicine (hereafter referred to as *E. cava*) for centuries because of its various health benefits, including anti-inflammatory and antioxidant properties [28]. Recent studies have revealed that *E. cava* contains various polyphenol components and other bioactive compounds that may affect the CNS [29,30,31]. Therefore, we expected that *E. cava* extract would regulate NI induced by chronic inflammatory conditions.

Surprisingly, the levels of markers related to inflammation and neurodegeneration were reduced in the brains of mice with chronic inflammatory conditions following treatment with *E. cava*. Although *E. cava* has been extensively studied, its role in controlling inflammation and preventing ND associated with chronic neuropathy is still unclear. This study focused on the effects of *E. cava* extract on NI and neurodegeneration.

## 2. Materials and Methods

### 2.1. Ecklonia cava Extract

*Ecklonia cava* (Amicogen, Inc., Jinju, Gyengsangnma-do, Republic of Korea) was bought in this experiment. Dried *E. cava* (1.0 kg) was extracted with water for 3~4 h with refluxing at 101 ± 1 °C and then filtered. To obtain *E. cava* powder, the extract was vacuum concentrated and lyophilized in a freeze-dryer. A quantitative analysis of the compounds of *E. cava* was shown in a previous study [32].

### 2.2. Animal Model

Male Crl:CD-1 (ICR) mice (6 weeks old) were obtained from Orient Bio Inc. (Seongnam, Republic of Korea). All of the animals were used in this study and housed in a pathogen-free facility at Chungnam National University. The mice were fed standard chow (5L79, Orient Bio Inc., Seongnam, Republic of Korea) with water provided ad libitum. All of the animal experiments were performed in accordance with the Chungnam Facility Animal Care Committee (202209A-CNU-191). The mice (*n* = 25) were divided into five groups (*n* = 5 control group, *n* = 5 lipopolysaccharide (LPS) group, *n* = 5 LPS + *E. cava* 10 mg/kg group, *n* = 5 LPS + *E. cava* 50 mg/kg group, *n* = 5 LPS + *E. cava* 100 mg/kg group). The mice had a 2 week acclimation period. For 25 days they were injected orally and the control group was treated with the same volume of distilled water as the *E. cava* extract treatment. On day 19, LPS (Lipopolysaccharides from Escherichia coli O111:B4, Sigma-Aldrich, St. Louis, MO, USA) was treated at 750 ug/kg/bodyweight for a week. Eight hours before the sacrifice, mice were treated with LPS at 1.5 mg/kg/body weight.

### 2.3. Western Blotting

Tissue protein samples from the cerebrum and hippocampus were prepared using a protein lysis buffer (78510, Thermo Fisher Scientific, Waltham, MA, USA) and measured by Bradford assay using a PRO-Measure solution (#21011, Intron, Kirkland, WA, USA). In addition, the serum sample was diluted with PBS and the experiment was conducted. Depending on the target protein size, the samples were run using 8% or 10% polyacrylamide gels and transferred to the PVDF membrane. After blocking for 1 h with 3% BSA100 (9048-46-8, LPS solution, Daejeon, Republic of Korea), the membranes were diluted TBS-T buffer (04870517TBST4021, LPS solution). Primary antibodies were incubated overnight at 4 °C. Sequentially, the membranes were washed with TBS-T, and the secondary antibodies were incubated for 90 min at room temperature. After that, the membrane was washed 3 times with TBS-T and then reacted with ECL solution (XLS025-0000, Cyanagen, Bologna, Italy) to detect the result with Chemi Doc (Fusion Solo, VilberLourmat, Collégien, France). The primary and secondary antibody information is shown in Table 1.

### 2.4. Total RNA Extraction and Real-Time Quantitative PCR

Using TRIzol reagents (15596-026, Life technologies, Carlsbad, CA, USA), total RNA was extracted in the cerebrum and hippocampus. cDNA was generated according to the protocol of the reverse transcription kit (SG-cDNAS100, Smartgene, Daejeon, Republic of Korea). Quantitative PCR (Stratagene Mx3000P, Agilent Technologies, Santa Clara, CA, USA) was performed using SYBR green SG-SYBR-500, Smartgene, Raleigh, NC, USA). The transcripts levels were calculated based on the cycle threshold and monitored for the amplification curve. RPLP0 was used for internal control. The primer sequences are listed in Table 2.

### 2.5. Immunohistochemistry

For immunohistochemistry staining, paraffin-embedded tissues were cut by 4μm and attached to silane coated slide. With serial hydration step in xylene, EtOH, and distilled water, and then progressed antigen retrieval step. The tissue slides were blocked with 3% BSA (9048-46-8, LPS solution, Daejeon, Republic of Korea) for 1 h. Primary antibodies were incubated overnight at 4 °C. After 3 times washing with TBS-T, the Alexa-Fluor secondary antibodies (#A21471, #A21207, Thermo Fisher Scientific Inc., Waltham, MA, USA) were incubated at room temperature. After 3 times washing, the slides were mounted in ProLong Gold antifade reagent (P36935, Thermo Fisher Scientific Inc., Waltham, MA, USA). The slides were examined using a DMi8 microscope (Leica Microsystems, Wetzlar, Germany).

### 2.6. Statistical Analysis

Data are reported as an average ± SEM. The differences between means were obtained through one-way ANOVA, and then a Tukey post-analysis was performed using Graph Pad Software (GraphPad Inc., San Diego, CA, USA).

## 3. Results

### 3.1. E. cava Reduces Pro-Inflammatory Cytokines Induced by Lipopolysaccharide

ICR mice were exposed to *E. cava* for 25 days and then exposed to lipopolysaccharide (LPS) for 7 days. The mouse brain tissue was sampled and divided into the cerebrum and hippocampus for analysis (Figure 1A). The body weights of the mice were monitored before euthanasia (Figure 1B). Compared with the control group, the LPS group (88%), LPS + *E. cava* 10 mg/kg group (87%), LPS + *E. cava* 50 mg/kg group (85%), and LPS + *E. cava* 100 mg/kg group (87%) showed decreased weight (*p* < 0.05, Figure 1B). However, no differences were observed between the groups treated with LPS. When we measured pro-inflammatory cytokines, we found that *E. cava* regulated chronic inflammation induced by LPS. The levels of serum interleukin-1 beta (IL-1β) increased significantly (*p* < 0.001, 2.75-fold) in the LPS group compared to the control group (Figure 1C). The LPS + *E. cava* 10 mg/kg group did not differ from the LPS group. However, the LPS + *E. cava* 50 mg/kg and LPS + *E. cava* 100 mg/kg groups showed a significant decrease (*p* < 0.001, 93%, and 80%, respectively) compared to the LPS group (Figure 1C). Furthermore, we measured IL-1β and interleukin-6 (IL-6) mRNA levels in the cerebrum and hippocampus. In the cerebrum, IL-1β mRNA levels increased significantly (*p* < 0.001, 13.5-fold) in the LPS group compared to the control group, and the LPS + *E. cava* 10 mg/kg group (41%), the LPS + *E. cava* 50 mg/kg group (8%) and the LPS + *E. cava* 100 mg/kg group (14%) showed a significant decrease (*p* < 0.001) compared to the LPS group (Figure 1D). IL-6 mRNA levels increased significantly (*p* < 0.001, 14.5-fold) in the LPS group compared to the control group, and the LPS + *E. cava* 10 mg/kg (50%), LPS + *E. cava* 50 mg/kg (39%), and LPS + *E. cava* 100 mg/kg (26%) groups showed a dose-dependent decrease (*p* < 0.001) compared to the LPS group (Figure 1D). In the hippocampus, IL-1β mRNA levels increased significantly (*p* < 0.001, 14.9-fold) in the LPS group compared to the control group and the LPS + *E. cava* 10 mg/kg (76%), the LPS + *E. cava* 50 mg/kg (14%) and the LPS + *E. cava* 100 mg/kg (36%) groups showed a significant decrease (*p* < 0.001) compared to the LPS group (Figure 1E). At the same time, the differences in IL-6 mRNA levels were not significant in all groups.

### 3.2. E. cava Decreases LPS-Induced NF-κB and STAT3 Activation in Mouse Cerebrum

Based on previous results, pro-inflammatory cytokines have been shown to be induced by LPS. Furthermore, *E. cava* decreases the secretion of pro-inflammatory cytokines. Accordingly, we measured the activation of the nuclear factor kappa-light-chain-enhancer of activated B-cells (NF-κB) and signal transducer and activator of transcription 3 (STAT3) genes, which regulate the immune response and inflammation. In the cerebrum, the phospho-nuclear factor of kappa light polypeptide gene enhancer in B-cells inhibitor, alpha (IκBα) protein level increased significantly (*p* < 0.001, 2.89-fold) in the LPS group compared to the control group. The LPS + *E. cava* 50 mg/kg group (65%) and the LPS + *E. cava* 100 mg/kg group (48%) showed a significant decrease (*p* < 0.001) in the phospho-IκBα protein level compared to the LPS group (Figure 2). IκBα protein levels did not differ significantly among all groups. However, the pIκBα/IκBα ratio increased significantly (0.001, 3.11-fold) in the LPS group compared to the control group. The LPS + *E. cava* 50 mg/kg group (55%) and the LPS + *E. cava* 100 mg/kg group (43%) showed a significant decrease compared to the LPS group (Figure 2). Next, the phospho-NF-κB protein level increased significantly (*p* < 0.001, 3.28-fold) in the LPS group compared to the control group. The LPS + *E. cava* 50 mg/kg group (78%) and the LPS + *E. cava* 100 mg/kg group (48%) showed a significant decrease (*p* < 0.001) in the phospho-NF-κB protein level compared to the LPS group (Figure 2). NF-κB protein levels did not differ significantly among all groups. However, the pNF-κB/NF-κB ratio increased significantly (*p* < 0.001, 3.73-fold) in the LPS group compared to the control group. The LPS + *E. cava* 50 mg/kg group (*p* < 0.05, 74%) and the LPS + *E. cava* 100 mg/kg group (*p* < 0.001, 36%) showed a significant decrease compared to the LPS group (Figure 2). The phospho-STAT3 protein level increased significantly (*p* < 0.05, 1.45-fold) in the LPS group compared to the control group. The LPS + *E. cava* 10 mg/kg (*p* < 0.05, 82%), LPS + *E. cava* 50 mg/kg (*p* < 0.001, 65%), and LPS + *E. cava* 100 mg/kg (*p* < 0.001, 55%) groups showed a significant decrease in phospho-STAT3 protein levels compared to the LPS group (Figure 2). The STAT3 protein levels did not differ significantly between the control and LPS groups. However, the LPS + *E. cava* 10 mg/kg (1.42-fold), LPS + *E. cava* 50 mg/kg (1.6-fold), and LPS + *E. cava* 100 mg/kg (1.68-fold) groups showed a dose-dependent increase (*p* < 0.001) compared to the LPS group (Figure 2). Additionally, pSTAT3/STAT3 levels increased significantly (*p* < 0.001, 1.31-fold) in the LPS group compared to those in the control group. The LPS + *E. cava* 10 mg/kg (58%), LPS + *E. cava* 50 mg/kg (41%), and LPS + *E. cava* 100 mg/kg (34%) groups showed a dose-dependent decrease (*p* < 0.001) compared to the LPS group (Figure 2).

### 3.3. E. cava Decreases LPS-Induced NF-κB and STAT3 Activation in Mouse Hippocampus

In the hippocampus, the phospho-IκBα protein level increased significantly (*p* < 0.001, 2.15-fold) in the LPS group compared to the control group. The LPS + *E. cava* 10 mg/kg (*p* < 0.05, 85%), LPS + *E. cava* 50 mg/kg (*p* < 0.001, 66%), and LPS + *E. cava* 100 mg/kg (*p* < 0.001, 60%) groups showed a dose-dependent decrease compared to the LPS group (Figure 3). IκBα protein levels did not differ significantly among all groups. However, the pIκBα/IκBα ratio increased significantly (*p* < 0.001, 2.15-fold) in the LPS group compared to the control group. The LPS + *E. cava* 10 mg/kg (*p* < 0.05, 84%), LPS + *E. cava* 50 mg/kg (*p* < 0.001, 68%) group, and the LPS + *E. cava* 100 mg/kg (*p* < 0.001, 65%) groups showed a dose-dependent decrease compared to the LPS group (Figure 3). Next, the phospho-NF-κB protein level increased significantly (*p* < 0.001, 1.53-fold) in the LPS group compared to the control group. The LPS + *E. cava* 50 mg/kg (*p* < 0.001, 61%) group and the LPS + *E. cava* 100 mg/kg (*p* < 0.001, 47%) group showed a significant decrease in the phospho-NF-κB protein level compared to the LPS group (Figure 3). NF-κB protein levels did not differ significantly among all groups. However, the pNF-κB/NF-κB ratio increased significantly (*p* < 0.001, 1.57-fold) in the LPS group compared to the control group. Additionally, The LPS + *E. cava* 50 mg/kg (*p* < 0.001, 60%) and LPS + *E. cava* 100 mg/kg (*p* < 0.001, 47%) groups showed a significant decrease compared to the LPS group (Figure 3). The phospho-STAT3 protein level increased significantly (*p* < 0.001, 1.83-fold) in the LPS group compared to the control group. The LPS + *E. cava* 10 mg/kg (*p* < 0.01, 78%), LPS + *E. cava* 50 mg/kg (*p* < 0.01, 77%), and LPS + *E. cava* 100 mg/kg (*p* < 0.01, 78%) groups showed a significant decrease in phospho-STAT3 protein levels compared to the LPS group (Figure 3). STAT3 protein levels did not differ significantly between groups (Figure 3). Additionally, pSTAT3/STAT3 levels increased significantly (*p* < 0.001, 1.66-fold) in the LPS group compared to those in the control group. The LPS + *E. cava* 10 mg/kg (*p* < 0.01, 69%), LPS + *E. cava* 50 mg/kg (*p* < 0.01, 72%), and LPS + *E. cava* 100 mg/kg (*p* < 0.05, 79%) groups showed a significant decrease (*p* < 0.05) compared to the LPS group (Figure 3).

### 3.4. E. cava Reduces Apoptosis Caused by Neuroinflammation

We observed that factors related to inflammatory control increased in response to LPS and that these factors are known to be involved in apoptosis [33]. In addition, apoptosis mediated by caspase-3 is linked to brain damage [34]. We expected that apoptosis would decrease because *E. cava* extract reduces inflammation.

In the cerebrum, the cleaved caspase-3 protein level increased significantly (*p* < 0.001, 2.52-fold) in the LPS group compared to the control group. The LPS + *E. cava* 50 mg/kg (58%) and LPS + *E. cava* 100 mg/kg (51%) groups showed a significant decrease (*p* < 0.001) in cleaved caspase-3 protein level compared with the LPS group (Figure 4A). The caspase-3 protein levels did not differ significantly between the groups. However, the cleaved caspase-3 to caspase-3 ratio was increased significantly (*p* < 0.001, 2.82-fold) in the LPS group compared to those in the control group. Additionally, The LPS + *E. cava* 50 mg/kg (56%) and LPS + *E. cava* 100 mg/kg (46%) groups showed a significant decrease (*p* < 0.001) compared to the LPS group (Figure 4A).

In the hippocampus, the cleaved caspase-3 protein level increased significantly (*p* < 0.001, 3.39-fold) in the LPS group compared to the control group. The LPS + *E. cava* 50 mg/kg (*p* < 0.01, 86%) and LPS + *E. cava* 100 mg/kg (*p* < 0.001, 81%) groups showed a significant decrease in cleaved caspase-3 protein level compared with the LPS group (Figure 4B). The caspase-3 protein levels did not differ significantly between the groups. However, the cleaved caspase-3 to caspase-3 ratio was increased significantly (*p* < 0.001, 3.47-fold) in the LPS group compared to the control group. Additionally, The LPS + *E. cava* 50 mg/kg (90%) and LPS + *E. cava* 100 mg/kg (90%) groups showed a significant decrease (*p* < 0.05) compared to the LPS group (Figure 3).

### 3.5. E. cava Reduces Glial Fibrillary Acidic Protein (GFAP) Expression in Mouse Brain

Astrocytes are known to regulate the immune response in the damaged CNS, and brain damage, such as injury, inflammation, and disease, increases the expression of GFAP proteins in astrocytes [35]. Therefore, we measured the expression of GFAP, a known astrocyte marker. When astrocytes in the cerebrum were stained with the GFAP antibody, positive signals were higher (*p* < 0.001, 1.70-fold) in the LPS group than in the control group. Additionally, a concentration-dependent decrease was observed in the LPS + *E. cava* 10 mg/kg (*p* < 0.001, 78%), LPS + *E. cava* 50 mg/kg (*p* < 0.001, 63%), and LPS + *E. cava* 100 mg/kg (*p* < 0.001, 55%) groups compared to that in the LPS group (Figure 5A). Furthermore, GFAP protein levels increased significantly (*p* < 0.001, 1.78-fold) in the LPS group compared to the control group, the LPS + *E. cava* 10 mg/kg (*p* < 0.05, 86%), LPS + *E. cava* 50 mg/kg (*p* < 0.05, 87%) and LPS + *E. cava* 100 mg/kg (*p* < 0.001, 64%) groups showed a significant decrease in GFAP protein levels compared to the LPS group (Figure 5B).

In the hippocampus, GFAP antibody, positive signals were higher (*p* < 0.001, 1.50-fold) in the LPS group than in the control group. The LPS + *E. cava* 50 mg/kg (72%) and LPS + *E. cava* 100 mg/kg (65%) groups showed a significant decrease (*p* < 0.001) in GFAP quantitative level compared with the LPS group (Figure 5C). GFAP protein levels increased significantly (*p* < 0.001, 1.65-fold) in the LPS group compared to the control group, the LPS + *E. cava* 50 mg/kg (*p* < 0.001 48%) and LPS + *E. cava* 100 mg/kg (*p* < 0.001, 50%) groups showed a significant decrease in GFAP protein levels compared to the LPS group (Figure 5D).

### 3.6. E. cava Weakens Alzheimer’s Markers by Chronic Neuroinflammation

Based on the previous results, we observed that chronic inflammatory responses lead to NI. NI is known to induce neurodegeneration [18,19]. Therefore, we measured markers related to Alzheimer’s disease. In the cerebrum, APP and tau protein levels did not differ significantly among the groups. While beta-amyloid protein levels increased significantly (*p* < 0.001, 3.07-fold) in the LPS group compared to the control group, the LPS + *E. cava* 50 mg/kg (*p* < 0.001, 53%) and LPS + *E. cava* 100 mg/kg (*p* < 0.001, 55%) groups showed a significant decrease in beta-amyloid protein levels compared to the LPS group (Figure 6A). In the hippocampus, tau protein levels did not differ significantly between groups. However, APP protein levels increased significantly (*p* < 0.001, 1.48-fold) in the LPS group compared to the control group, whereas the LPS + *E. cava* 50 mg/kg (*p* < 0.001, 54%) and LPS + *E. cava* 100 mg/kg (*p* < 0.001, 48%) groups showed a significant decrease in APP protein levels compared to the LPS group (Figure 6B). Additionally, beta-amyloid protein levels increased significantly (*p* < 0.001, 2.16-fold) in the LPS group compared to the control group, while the LPS + *E. cava* 50 mg/kg group (*p* < 0.01, 80%) and the LPS + *E. cava* 100 mg/kg (*p* < 0.001, 68%) group showed a significant decrease in beta-Amyloid protein levels compared to the LPS group (Figure 6B).

## 4. Discussion

Chronic inflammation is increasing worldwide, largely due to aging, lifestyle, and environmental factors [2]. Chronic inflammation can lead to chronic diseases, such as diabetes, obesity, cardiovascular diseases, and neurodegenerative diseases [2]. NDs are known to be mostly caused by inflammation. NI is considered chronic rather than acute [16,36]. NI is associated with ND [18,20,37] and reducing inflammation is expected to weaken the progression of ND. Strikingly, several studies have reported that inhibition of inflammatory cytokines reduces nerve loss in ND [38,39,40].

NI drugs include nonsteroidal anti-inflammatory drugs (NSAIDs), disease-modifying antirheumatic drugs (DMARDs), and corticosteroids. However, they have various side effects, such as gastrointestinal disorders, kidney damage, and osteoporosis [41,42,43]. Therefore, safe and effective treatment options must be identified. Recently, numerous studies have investigated the protective effects of natural products and bioactive compounds against neurological diseases [21,22,44]. Flavonoids, phlorotannins, curcumin, resveratrol, and genistein are among the compounds that have shown potential for treating and preventing NI [23,24,25,26,27]. Inflammation and oxidative stress play a role in neurodegenerative diseases. In vitro and in vivo studies have demonstrated that antioxidant plant polyphenols, such as resveratrol and curcumin, can reduce the deposition of amyloid-beta in Alzheimer’s disease, inhibit ROS production and cell death through their antioxidant effects, and demonstrate neuroprotective effects through anti-inflammatory and antioxidant mechanisms in Parkinson’s disease. It is expected that the neuroprotective actions of polyphenols in neurodegenerative diseases involve the downregulation of pro-inflammatory transcription factors, such as NF-kB, through their anti-inflammatory effects in the brain [45,46]. Phlorotannin is a polyphenol that is specifically present in brown algae and in the natural product *E. cava* [28]. Some studies have suggested that *E. cava* has antioxidant, anti-inflammatory, immunomodulatory, and neuroprotective effects [29,30,31]. Based on this evidence, we observed that *E. cava* reduced NI and neurodegeneration in mice with LPS-induced chronic inflammation.

Chronic inflammation is known to result in the secretion of cytokines that regulate the immune response and hematopoietic action by acting on cells involved in host defense and damage healing [1,5,47]. We measured IL-1β, the brain’s main inflammatory cytokine [11,48], in the serum. The LPS group showed an increase in IL-1β levels compared to the control group, and the groups that were administered *E. cava* showed a decrease in IL-1β levels in both the LPS + *E. cava* 50 mg/mL group and the LPS + *E. cava* 100 mg/mL group compared to the LPS group. In addition, we measured mRNA levels of IL-1β and IL-6 in the cerebrum and hippocampus. In the cerebrum, the LPS group showed increased levels of both IL-1β and IL-6 mRNA compared to the control group, while all groups administered LPS + *E. cava* showed lower levels than the LPS group. In the hippocampus, the LPS group showed increased levels of IL-1β mRNA compared to the control group, and LPS + *E. cava* groups administered *E. cava* showed lower levels than the LPS group. Overall, our results suggest that chronic inflammation induces NI through LPS and that *E. cava* may regulate cytokines through its anti-inflammatory properties.

Furthermore, many studies have shown that interleukin (IL) 1 beta and IL-6 signaling is activated by NF-kB and STAT3 when inflammation occurs through LPS, and NF-kB and STAT3 are also known to interact with cytokines and are involved in regulating the immune response and inflammation [49,50,51]. Therefore, we measured the activation of NF-κB and STAT3 to investigate the potential mechanisms underlying the anti-inflammatory effects of *E. cava*. In the cerebrum, LPS increased levels of pIκBα, pNF-κB, and pSTAT3 proteins compared to the control group. In groups administered LPS + *E. cava*, pIκBα and pNF-κB levels were decreased compared to the LPS group at concentrations of 50 mg/mL and 100 mg/mL, and the level of pSTAT3 protein decreased in a concentration-dependent manner compared to the LPS group. Furthermore, in the hippocampus, the LPS group had increased levels of pIκBα, pNF-κB, and pSTAT3 proteins compared to the control group. In groups administered LPS + *E. cava*, pNF-κB levels were decreased compared to the LPS group at concentrations of 50 mg/mL and 100 mg/mL, and the levels of pIκBα and pSTAT3 proteins decreased in a concentration-dependent manner compared to the LPS group. Our results suggest that *E. cava* treatment downregulates the phosphorylation of IκBα, NF-κB, and STAT3 in NI brains, demonstrating anti-inflammatory effects.

Our results have shown that cytokines and inflammatory response-related proteins are increased in NI states and that *E. cava* decreases levels of cytokines and inflammatory regulatory proteins. In addition, chronic inflammatory animal models are known to display persistent inflammatory reactions [52]. Continuous inflammatory reactions promote apoptosis [1], and activation of NF-κB [53,54,55,56] and STAT3 [57,58,59,60] is known to induce apoptosis in various organs as well as the brain. Therefore, we measured the capase-3 protein, an indicator of apoptosis, in a chronic NI state. Interestingly, the group treated with LPS showed a higher ratio of cleaved caspase-3 to capase-3 in the cerebrum and hippocampus than the control group. Moreover, the LPS + *E. cava* 50 and 100 mg/mL groups showed decreased protein levels of cleaved caspase-3 compared to the LPS group. We suggest that *E. cava* might exert anti-apoptotic effects that protect nerve cells from death. Neuronal cell death is one of the main pathological features of ND. Therefore, we expect *E. cava* to help with NI as well as ND.

Since *E. cava* has shown a protective effect against NI and apoptosis, and given that NI and ND are interrelated processes, we measured glial fibrillary acidic protein (GFAP), a biomarker used to diagnose Alzheimer’s disease [61]. Quantitative levels of GFAP in the cerebrum were higher in the LPS group than in the control group. Interestingly, GFAP IHC staining decreased in a concentration-dependent manner in the LPS + *E. cava* group. Additionally, we measured the levels of Alzheimer’s disease-related markers. Interestingly, in the LPS group, amyloid-beta protein levels increased in the cerebrum, whereas both amyloid-beta and APP protein levels increased in the hippocampus, compared to those in the control group. In the LPS group, amyloid-beta levels increased in the cerebrum, while both amyloid-beta and APP protein levels increased in the hippocampus compared to the control group, and compared to the LPS group, it was reduced in LPS + *E. cava* 50 mg/mL and 100 mg/mL groups. These results suggested that *E. cava* reduces the risk of developing ND caused by chronic NI. Thus, *E.cava* was considered to improve NI and ND. In summary, we have shown that *E. cava* dampen the LPS-mediated NF-kB and STAT3 activation, leading to the signaling related to inflammation and immune responses in the brain. As a natural product, *E. cava* contains phlorotannins which are polyphenols of marine algae. Phlorotannin-rich *E. cava* has the potential for functional foods with antioxidant and anti-inflammatory activities [62]. One of the major polyphenolic compounds in *E. cava*, Dieckol, may also play a key role in allergic inflammatory reactions [63]. By reducing NF-kB and STAT3 activation, we suggest that *E. cava* could protect against apoptosis and neurodegeneration. Based on this rationale, this study provides evidence that *E. cava* reduce neuroinflammation and play a substance in neurodegenerative disease therapeutics.

## 5. Conclusions

To summarize our results, we induced chronic inflammation using LPS to activate cytokine- and inflammation-related genes, leading to NI. We showed that *E. cava* has anti-inflammatory and anti-apoptotic effects under NI conditions. Moreover, *E. cava* may exert protective effects against ND. These findings suggest that *E. cava* can be used to treat chronic NI and ND.

## Figures and Tables

**Figure 1 nutrients-15-02007-f001:**
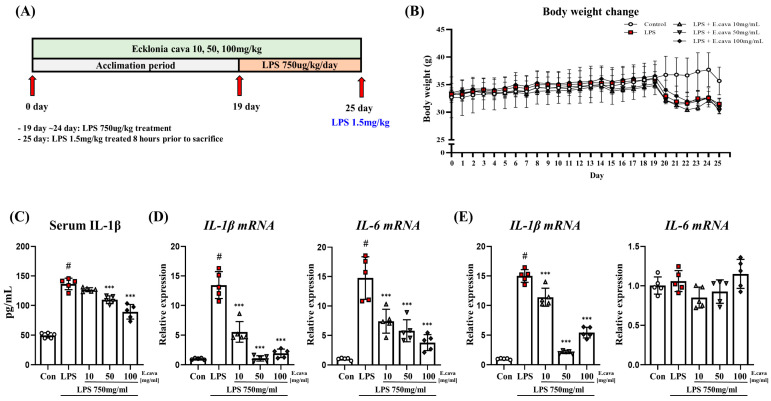
*E. cava* decreases pro-inflammatory cytokines levels. (**A**) The schematic diagram displays the schedule of animal experiments. The mice (*n* = 25) were divided into five groups (*n* = 5 control group, *n* = 5 Lipopolysaccharide (LPS) group, *n* = 5 LPS + *E. cava* 10 mg/kg group, *n* = 5 LPS + *E. cava* 50 mg/kg group, *n* = 5 LPS + *E. cava* 100 mg/kg group). For 25 days, they were injected orally, and the control group was treated with the same capacity of distilled water, which was used for the *E. cava* extract treatment. On day 19, LPS was treated at 750 ug/kg/body weight for a week. The mice were sacrificed before 8 h LPS treatment at 1.5 mg/kg/bodyweight, and their organs were isolated. (**B**) Change of mouse body weight (**C**) Serum blood IL-1β level measured after sacrifice. (**D**) QRT-PCR measured IL-1β and IL-16 genes mRNA levels in the cerebrum of each group of male mice. (**E**) QRT-PCR measured IL-1β and IL-16 genes mRNA levels in the hippocampus. RPLP0 was used for internal control. The points on the graph are represented by different shapes, with a white circle indicating the control group, a red square indicating the LPS group, a triangle representing the LPS + *E. cava* 10 mg/kg group, an inverted triangle indicating the LPS + *E. cava* 50 mg/kg group, and a rhombus representing the LPS + *E. cava* 100 mg/kg group. The differences between means were obtained through one-way ANOVA, and then a Tukey post-analysis. The values represent means ± S.D. # *p* < 0.001 vs. control group and *** *p* < 0.01 vs. LPS group.

**Figure 2 nutrients-15-02007-f002:**
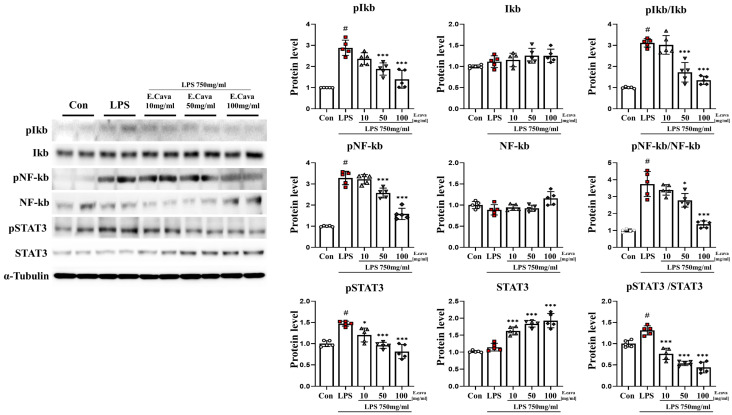
*E. cava* reduces inflammation-regulated markers in the cerebrum. Western blot analysis and quantification of inflammation-regulated related genes were evaluated in the cerebrum of each group of male mice. Alpha-Tubulin was used for internal control. The points on the graph are represented by different shapes, with a white circle indicating the control group, a red square indicating the LPS group, a triangle representing the LPS + *E. cava* 10 mg/kg group, an inverted triangle indicating the LPS + *E. cava* 50 mg/kg group, and a rhombus representing the LPS + *E. cava* 100 mg/kg group. The differences between means were obtained through one-way ANOVA, and then a Tukey post-analysis. The values represent means ± S.D. # *p* < 0.001 vs. control group and * *p* < 0.05, *** *p* < 0.01 vs. LPS group.

**Figure 3 nutrients-15-02007-f003:**
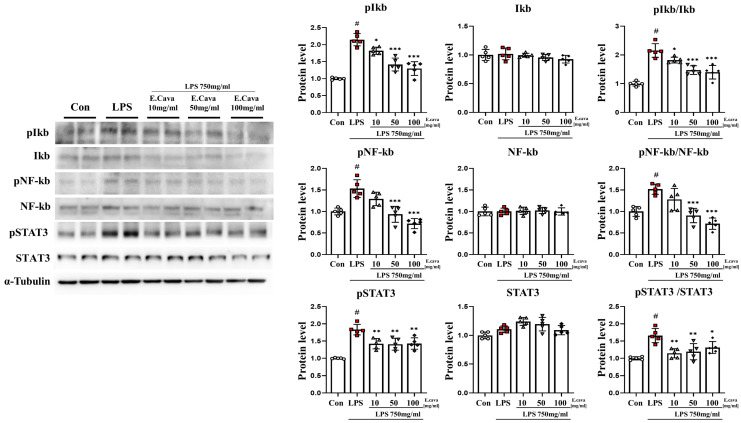
*E. cava* reduces inflammation-regulated markers in the hippocampus. Western blot analysis and quantification of inflammation-regulated related genes were evaluated in the hippocampus of each group of male mice. Alpha-Tubulin was used for internal control. The points on the graph are represented by different shapes, with a white circle indicating the control group, a red square indicating the LPS group, a triangle representing the LPS + *E. cava* 10 mg/kg group, an inverted triangle indicating the LPS + *E. cava* 50 mg/kg group, and a rhombus representing the LPS + *E. cava* 100 mg/kg group. The differences between means were obtained through one-way ANOVA, and then a Tukey post-analysis. The values represent means ± S.D. # *p* < 0.001 vs. control group and * *p* < 0.05 ** *p* < 0.01, *** *p* < 0.01 vs. LPS group.

**Figure 4 nutrients-15-02007-f004:**
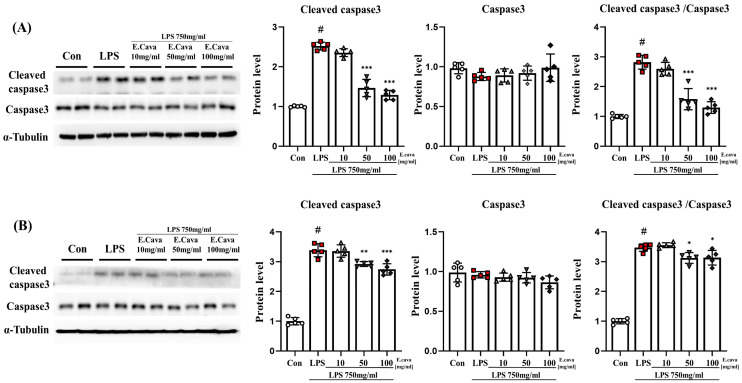
*E. cava* reduces apoptosis markers in mouse brains. (**A**) Western blot analysis and quantification of apoptosis-related genes were evaluated in the cerebrum of each group of male mice. (**B**) Western blot analysis and quantification of apoptosis-related genes were evaluated in the hippocampus of each group of male mice. Alpha-Tubulin was used for internal control. The points on the graph are represented by different shapes, with a white circle indicating the control group, a red square indicating the LPS group, a triangle representing the LPS + *E. cava* 10 mg/kg group, an inverted triangle indicating the LPS + *E. cava* 50 mg/kg group, and a rhombus representing the LPS + *E. cava* 100 mg/kg group. The differences between means were obtained through one-way ANOVA, and then a Tukey post-analysis. The values represent means ± S.D. # *p* < 0.001 vs. control group and * *p* < 0.05 ** *p* < 0.01, *** *p* < 0.01 vs. LPS group.

**Figure 5 nutrients-15-02007-f005:**
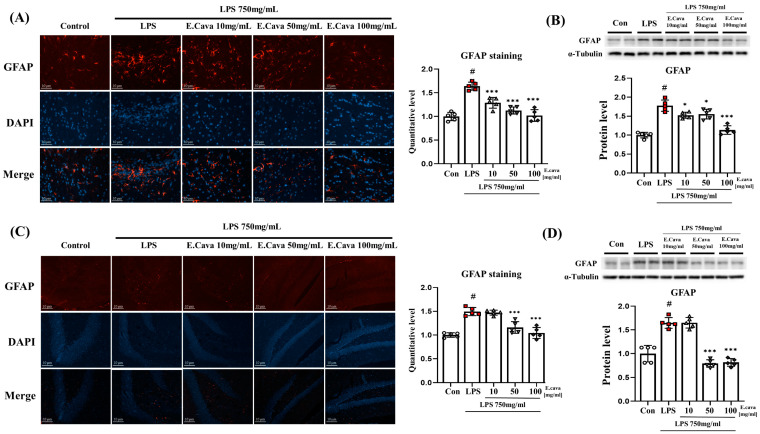
*E. cava* reduces glial fibrillary acidic protein (GFAP) expression in mouse brain (**A**) Astrocyte cells are characterized in GFAP staining (Scale bar, 10 µm). Representative immunohistochemistry (IHC) of cerebrum astrocytes in the mouse brain indicates GFAP (red) and dapi (blue). Quantification of GFAP staining was analyzed by Image J as setting white holes for positive standards. (**B**) Western blot analysis and quantification of GFAP were evaluated in the cerebrum of each group of male mice. Alpha-Tubulin was used for internal control. (**C**) Representative immunohistochemistry (IHC) of hippocampus astrocytes in mouse brain indicates GFAP (red) and dapi (blue). Quantification of GFAP staining was analyzed by Image J as setting white holes for positive standards. (**D**) Western blot analysis and quantification of GFAP were evaluated in the hippocampus of each group of male mice. Alpha-Tubulin was used for internal control. The points on the graph are represented by different shapes, with a white circle indicating the control group, a red square indicating the LPS group, a triangle representing the LPS + *E. cava* 10 mg/kg group, an inverted triangle indicating the LPS + *E. cava* 50 mg/kg group, and a rhombus representing the LPS + *E. cava* 100 mg/kg group. The differences between means were obtained through one-way ANOVA, and then a Tukey post-analysis. The values represent means ± S.D. # *p* < 0.001 vs. control group and * *p* < 0.05, *** *p* < 0.01 vs. LPS group.

**Figure 6 nutrients-15-02007-f006:**
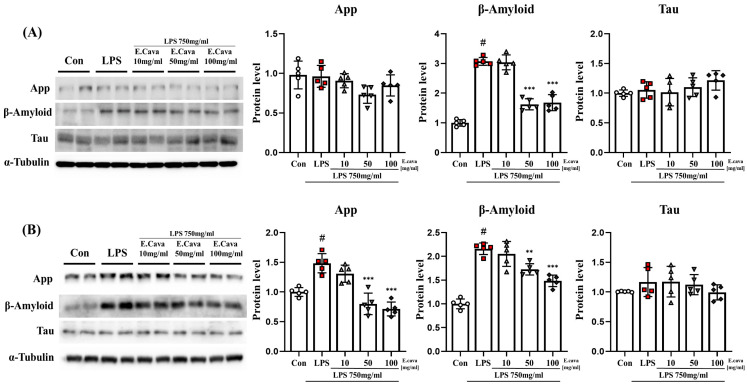
*E. cava* reduces Alzheimer’s disease-related markers in the mouse brain. (**A**) Western blot analysis and quantification of neurodegenerative disease-related genes were evaluated in the cerebrum of each group of male mice. (**B**) Western blot analysis and quantification of Alzheimer’s disease-related genes were evaluated in the hippocampus of each group of male mice. Alpha-Tubulin was used for internal control. The points on the graph are represented by different shapes, with a white circle indicating the control group, a red square indicating the LPS group, a triangle representing the LPS + *E. cava* 10 mg/kg group, an inverted triangle indicating the LPS + *E. cava* 50 mg/kg group, and a rhombus representing the LPS + *E. cava* 100 mg/kg group. The differences between means were obtained through one-way ANOVA, and then a Tukey post-analysis. The values represent means ± S.D. # *p* < 0.001 vs. control group and ** *p* < 0.01, *** *p* < 0.01 vs. LPS group.

**Table 1 nutrients-15-02007-t001:** Primary and secondary antibodies.

Primary Antibodies	Type	Lot.	Inc.
phospho-IκBα	Rabbit monoclonal	#2697	Cell signaling technology
IκBα	Mouse monoclonal	#4814	Cell signaling technology
phospho-NF-κB	Rabbit monoclonal	#3033	Cell signaling technology
NF-κB	Rabbit monoclonal	#8242	Cell signaling technology
phospho-STAT3	Rabbit monoclonal	AP0070	Company ABclonal, Inc.
STAT3	Mouse monoclonal	A1192	Company ABclonal, Inc.
Cleaved caspase3	Rabbit monoclonal	#9664	Cell signaling technology
Caspase3	Rabbit monoclonal	#9665	Cell signaling technology
GFAP	Rabbit monoclonal	A19058	Company ABclonal, Inc.
Amyloid-beta	Mouse monoclonal	sc-28365	Santa Cruz biotechology
Tau	Rabbit monoclonal	A1103	Company ABclonal, Inc.
Secondary antibodies	Type	Lot.	Inc.
Anti-Mouse IgG	Goat	121507	Jackonimmuno
Anti-Rabbit IgG	Mouse	123213	Jackonimmuno

**Table 2 nutrients-15-02007-t002:** Primers.

Gene	Forward Primer (5′-3′)	Reward Primer (5′-3′)	Species
*IL-1β*	GCC CAT CCT CTG TGA CTC AT	AGG CCA CAG GTA TTT TGT CG	Mouse
*IL-6*	AGT TGC CTT CTT GGG ACT GA	TCC ACG ATT TCC CAG AGA AC	Mouse
*RPLP0*	GCA GCA GAT CCG CAT GTC GCT CCG	GAG CTG GCA CAG TGA CCT CAC ACG G	Mouse

## Data Availability

Not applicable.

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
