# Peer review of "Neuroprotective Effects of Ecklonia cava in a Chronic Neuroinflammatory Disease Model"

_nutrients, 2023, doi:10.3390/nu15082007_

Round 1
Reviewer 1 Report
The author explore the effect E. cava in neuroinflammation-induced neurodegenerative disease. Some issues need to be clarified.
1. How the author decide the time points and therapy duration of E.cave and LPS.
2. The author mentioned "The mice (n = 30) were divided into five groups", while "(n = 5 control group, n = 5, Lipopolysaccharide (LPS) group, n = 5 LPS + E. cava 10mg/kg group, n = 5 LPS + E. cava 50mg/kg group, n = 5 LPS + E. cava 100mg/kg group" conclude a total number of 25 mice.
3. References list are missing in the manuscript.
4. Figure 1D and E: The trend of IL-6 mRNA in cerebrum and hippocampus are significant different. Please explain the possible reason.
5. The background of WB are obscure, could the authors provide the original uncropped WB gel for further check.
6. How do the author decide the dosage of E.cova in this study?
7. Figure 5, scale bar need to be added.
8. Figure 5, it seems GFAP could be completely reversed to normal level in E.Cava 100mg group. Could the author add the results of GFAP by WB method.
Author Response
Reviewer 1
The author explore the effect E. cava in neuroinflammation-induced neurodegenerative disease. Some issues need to be clarified.
1.How the author decide the time points and therapy duration of E.cave and LPS.
In our previous studies, we evaluated the effect of E. Cava extract in Polycystic Ovary Syndrome rats (PMID: 35928256 & PMID: 30524282). To introduce into the mouse model, we modified the concentration and exposed three different concentrations. To decide the time point and duration of LPS treatment, we have consulted the references (PMID: 30962497), and decided the concentration of LPS via preliminary study.
- The author mentioned "The mice (n = 30) were divided into five groups", while "(n = 5 control group, n = 5, Lipopolysaccharide (LPS) group, n = 5 LPS + E. cava 10mg/kg group, n = 5 LPS + E. cava 50mg/kg group, n = 5 LPS + E. cava 100mg/kg group" conclude a total number of 25 mice.
According to the reviewer’s indication, we revised the sentence.
(Line 89-92)
“The mice (n = 25) were divided into five groups (n = 5 control group, n = 5, Lipopolysaccharide (LPS) group, n = 5 LPS + E. cava 10mg/kg group, n = 5 LPS + E. cava 50mg/kg group, n = 5 LPS + E. cava 100mg/kg group).”
(Line167-170)
“Figure 1. E. cava decreases pro-inflammatory cytokines levels (A) The schematic diagram displays the schedule of animal experiments. The mice (n = 25) were divided into five groups (n = 5 control group, n = 5 Lipopolysaccharide (LPS) group, n = 5 LPS + E. cava 10mg/kg group, n = 5 LPS + E. cava 50mg/kg group, n = 5 LPS + E. cava 100mg/kg group).“
- References list are missing in the manuscript.
According to the reviewer’s indication, we added the reference. - Figure 1D and E: The trend of IL-6 mRNA in cerebrum and hippocampus are significant different. Please explain the possible reason.
The expression of mRNA can differ across different tissues due to various factors, including tissue-specific transcription factors, the number of glial cells, and the blood supply in the brain. These factors might influence the difference in IL-6 transcriptional levels. - The background of WB are obscure, could the authors provide the original uncropped WB gel for further check.
In original manuscript, we provided the original uncropped membrane and the indicated band from Western Blot as a PDF file. Please find the file. - How do the author decide the dosage of E.cova in this study?
In our previous studies, we evaluated the effect of E. Cava extract in Polycystic Ovary Syndrome rats (PMID: 35928256 & PMID: 30524282). To introduce into the mouse model, we modified the concentration and exposed three different concentrations. - Figure 5, scale bar need to be added.
According to the reviewer’s suggestion, we added the scale bar.
8. Figure 5, it seems GFAP could be completely reversed to normal level in E.Cava 100mg group. Could the author add the results of GFAP by WB method.
Following the reviewer's suggestion, we measured GFAP protein levels by Western blot and have included the resulting values, membrane figure, and quantitative graph in the manuscript.
"Please see the attachment"

Reviewer 2 Report
Figure 1 graphs C, D and E, should be adjusted, the error bars are confusing. Its seems unlikely that the larges differences and small differences in some of the bars would result in the same p value.
Abeta and tau are not markers of neurodegeneration as claimed in the text. Also make claim that E cava reduced neurodegeneration, but there is no neurodegeneration in this model nor any cell counting to support this claim.
Unclear why this study only used male mice. This should be addressed
Figure 5 needs to show n used in the study. Perhaps individual points like other graphs.
This statement should be referenced
“Previous studies have shown that cytokines and inflammatory response-related proteins are increased in NI states and that E. cava decreases levels of cytokines and inflammatory regulatory proteins.”
The current data shown does not exclude other cells types from having altered caspase effects too, so the following statement about neurons is a leap. “These findings suggest that E. cava exerts anti-apoptotic effects that protect nerve cells 379 from death.”
The discussion should not just rehash the results section, but could be expanded with some things like the following. It would be interesting to discuss how the authors believe there is a reduction in these signaling pathways and neuroinflammation. How does this relate to other polyphenol studies? Discuss success or failure of attempts to reduce inflammation in neurodegenerative diseases.
References were not present in pdf
Author Response
Reviewer 2
- Figure 1 graphs C, D and E, should be adjusted, the error bars are confusing. Its seems unlikely that the larges differences and small differences in some of the bars would result in the same p value.
Figures 1C, 1D, and 1E were adjusted based on the reviewer's comments, and the p-values were recalculated. Depending on the p-value, significance levels were adjusted as follows: P<0.001, * P<0.05, ** P<0.01, and *** P<0.001, and we modified the manuscript. - A beta and tau are not markers of neurodegeneration as claimed in the text. Also make claim that E cava reduced neurodegeneration, but there is no neurodegeneration in this model nor any cell counting to support this claim.
Amyloid beta (Abeta) and tau are indeed considered markers of neurodegeneration in the context of Alzheimer's disease [PMID: 31073121 & PMID: 28713158]. Abeta is a protein that forms plaques in the brain, and tau is a protein that forms tangles [PMID: 30635637]. The accumulation of Abeta and tau is associated with neuronal dysfunction and death, which are key features of neurodegeneration. These proteins are commonly used as biomarkers to diagnose and track the progression of Alzheimer's disease.
Reference
- “Diagnosis of Alzheimer’s disease utilizing amyloid and tau as fluid biomarkers” [PMID: 31073121]
- “Amyloid beta: structure, biology and structure-based therapeutic development” [PMID: 28713158]
- “ Tau PET imaging in neurodegenerative tauopathies—still a challenge” [PMID: 30635637]
- Unclear why this study only used male mice. This should be addressed
Female mice have a more complex hormonal cycle than males, which can cause variability in experimental results. This variability can make it more difficult to interpret data and draw accurate conclusions. - Figure 5 needs to show n used in the study. Perhaps individual points like other graphs.
According to the reviewer's opinion, we changed the graph of figure 5 to individual points graphs. - This statement should be referenced.
“Previous studies have shown that cytokines and inflammatory response-related proteins are increased in NI states and that E. cava decreases levels of cytokines and inflammatory regulatory proteins.”
The current data shown does not exclude other cells types from having altered caspase effects too, so the following statement about neurons is a leap. “These findings suggest that E. cava exerts anti-apoptotic effects that protect nerve cells 379 from death.”
At the beginning of the sentence, we revised the sentence. - The discussion should not just rehash the results section, but could be expanded with some things like the following. It would be interesting to discuss how the authors believe there is a reduction in these signaling pathways and neuroinflammation. How does this relate to other polyphenol studies? Discuss success or failure of attempts to reduce inflammation in neurodegenerative diseases.
According to the reviewer's opinion, we have discussed signaling pathways and neuroinflammation, the anti-inflammatory effects of polyphenols, as well as the success or failure of attempts to reduce inflammation in neurodegenerative diseases.
- References were not present in pdf
According to the reviewer’s suggestion, we added the reference.
"Please see the attachment."

Round 2
Reviewer 1 Report
The authors have addressed my concern.
Reviewer 2 Report
Response to second submission.
Figures have the p values correct but now does not state or show what the comparison is too. Are these values just compared to the control? It should state in the legend what statistics was used, eg ANOVA and was this done as a comparison to only the control or between all groups? The graphs are too confusing as they are.
I agree with the authors that amyloid beta and tau are markers of Alzheimer’s disease, but disagree that they are markers of neurodegeneration. You can have excessive amyloid plaques with out neurodegeneration this is an aspect of most APP expressing transgenic mice. There are also many instances of humans with large amounts of amyloid and no cognitive impairment. Neurodegeneration is by definition the loss of neurons. This manuscript at no point measures neuronal number or loss of neurons, and it is unlikely to occur in this model. Therefore statements that say E. cava prevents neurodegeneration are not correct. The title even reflects this statement and thus should be changed.
Yes female mice have more complex hormone cycles but this does not justify ignoring them. This aspect should be discussed as a flaw in the approach and that female animals should be studied in the future.
I do not think that the discussion was adequately improved. There is no mention of other polyphenol studies from the past, nor any mention of success or failures using polyphenols in neurodegenerative disease like AD.
Grammar in the figure legends needs some work.
Author Response
- Figures have the p values correct but now does not state or show what the comparison is too. Are these values just compared to the control? It should state in the legend what statistics was used, eg ANOVA and was this done as a comparison to only the control or between all groups? The graphs are too confusing as they are.
According to the reviewer's opinion, we revised figure legend. - I agree with the authors that amyloid beta and tau are markers of Alzheimer’s disease, but disagree that they are markers of neurodegeneration. You can have excessive amyloid plaques without neurodegeneration this is an aspect of most APP expressing transgenic mice. There are also many instances of humans with large amounts of amyloid and no cognitive impairment.
According to the reviewer's opinion, we revised the sentence.
(Line 338-339)
- cava weakens Alzheimer’s markers by chronic neuroinflammation
(Line 341-342)
Therefore, we measured markers related to Alzheimer’s disease.
(Line 357)
- cava reduces Alzheimer’s disease-related markers in mouse brain.
(Line 441-442)
Additionally, we measured the levels of Alzheimer’s disease-related markers.
- Neurodegeneration is by definition the loss of neurons. This manuscript at no point measures neuronal number or loss of neurons, and it is unlikely to occur in this model. Therefore statements that say E. cava prevents neurodegeneration are not correct. The title even reflects this statement and thus should be changed.
According to the reviewer's suggestion, we revised the title. - Yes female mice have more complex hormone cycles but this does not justify ignoring them. This aspect should be discussed as a flaw in the approach and that female animals should be studied in the future.
We used male mice to minimize variability in the experiment and observe the effect of E.cava. However, since the prevalence and physiology of neurodegenerative diseases are gender-specific, future studies should measure the effect of E.cava by inducing gender-specific neuroinflammation in both male and female mice. - I do not think that the discussion was adequately improved. There is no mention of other polyphenol studies from the past, nor any mention of success or failures using polyphenols in neurodegenerative disease like AD.
(Line 376-379)
We were mentioned polyphenol studies such as resveratrol, curcumin, and genistein. Polyphenols are a large group of naturally occurring compounds in plant foods, including fruits, vegetables, whole grains, tea, and wine. Resveratrol is found in the skin of red grapes, curcumin is the main ingredient in turmeric, and genistein is found in soybeans and other legumes. Also, According to the reviewer's opinion, we have discussed the successes and failures of using polyphenols in neurodegenerative diseases like AD. (line380-387)
Grammar in the figure legends needs some work.
According to the reviewer's opinion, we have confirmed the grammar of the figure legend.
"Please see the attachment."
